# PersonaAgent: When Large Language Model Agents Meet Personalization at Test Time

## Abstract

Large Language Model (LLM) empowered agents have recently emerged as advanced paradigms that exhibit impressive capabilities in a wide range of domains and tasks. Despite their potential, current LLM agents often adopt a one-size-fits-all approach, lacking the flexibility to respond to users' varying needs and preferences. This limitation motivates us to develop **PersonaAgent**, the first personalized LLM agent framework designed to address versatile personalization tasks. Specifically, PersonaAgent integrates two complementary components - a personalized memory module that includes episodic and semantic memory mechanisms; a personalized action module that enables the agent to perform tool actions tailored to the user. At the core, the *persona* (defined as unique system prompt for each user) functions as an intermediary: it leverages insights from personalized memory to control agent actions, while the outcomes of these actions in turn refine the memory. Based on the framework, we propose a **test-time user-preference alignment** strategy that simulate the latest $n$ interactions to optimize the *persona* prompt, ensuring real-time user preference alignment through textual loss feedback between simulated and ground-truth responses. Experimental evaluations demonstrate that PersonaAgent significantly outperforms other baseline methods by not only personalizing the action space effectively but also scaling during test-time real-world applications. These results underscore the feasibility and potential of our approach in delivering tailored, dynamic user experiences.

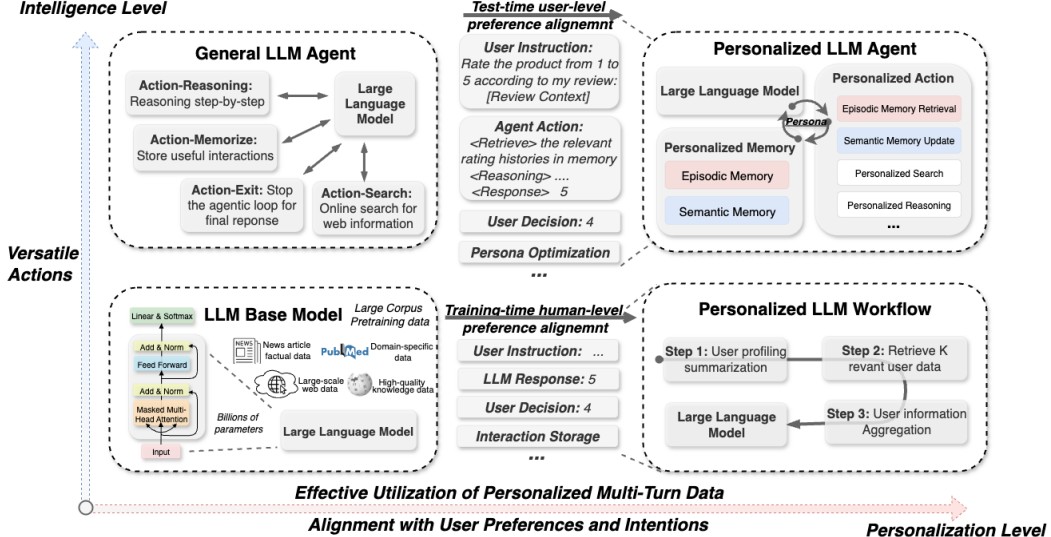

Figure 1: Design principles for personal intelligence and four representative frameworks. Note that personalized LLM agents perform personal-level alignment, whereas the others achieve only human-level alignment or merely store personal information.

# 1 INTRODUCTION

For a long time, humanity has pursued the ambitious goal of creating artificial intelligence capable of matching or surpassing human-level cognitive capabilities (Turing, 2009), thereby effectively assisting, augmenting, and enhancing human activities across numerous domains. This pursuit has been driven by two fundamental principles: achieving superior intelligence (Bubeck et al., 2023; Wooldridge & Jennings, 1995; Phan et al., 2025) and enhancing personalization (Rafieian & Yoganarasimhan, 2023; Kirk et al., 2024) as shown in Figure 1. Towards superior intelligence, large language models (LLMs), such as GPT (Achiam et al., 2023), Claude (Anthropic, 2024), and LLaMa (Touvron et al., 2023), have revolutionized various domains, demonstrating emergent capabilities in reasoning (Wei et al., 2022), language comprehension (Achiam et al., 2023), and instruction following (Ouyang et al., 2022). Beyond standalone LLMs, LLM-empowered agents (Luo et al., 2025) represent a paradigm shift, integrating external tools (Qin et al., 2024; Yuan et al., 2024; Wei et al., 2025), memory mechanisms (Hatalis et al., 2023; Zhong et al., 2024), and goal-directed reasoning (Yao et al., 2023b;a) to enhance their utility and autonomy. These agents move closer to human-like intelligence, capable of performing complex tasks and interacting with users more naturally and effectively. However, to truly harness the potential of these intelligent systems in everyday human contexts, it must capable of adapting tailored behaviors and interactions to cater to different users (Fischer, 2001). Despite their impressive versatility, existing LLMs and agents, primarily trained on generic large-scale datasets (Achiam et al., 2023; Anthropic, 2024; Touvron et al., 2023) or armed with general action tools (Yao et al., 2023b; Schick et al., 2023), inherently lack the capacity to dynamically utilize the user personal data and adapt to evolving preferences unique to each user.

Personalization, therefore, emerges as a critical factor for enabling agents to deliver more relevant responses, foster deeper user engagement, and establish trust through tailored interactions (Bickmore & Picard, 2005; Zhang et al., 2025a;b). As Table 1 highlights, achieving effective personalization intelligence can be measured from four critical perspectives: agentic intelligence, real-world applicability, personal data utilization, and preference alignment. Yet, balancing these dimensions simultaneously remains a fundamental challenge. Early efforts for aligning LLMs with human preferences, such as supervised fine-tuning (Zhang et al., 2023) and reinforcement learning from human feedback (RLHF) (Schulman et al., 2017; Rafailov et al., 2023), have improved the naturalness of instruction-following behaviors for generalized human preference but fall short in individual user preference alignment and personal data utilization. Recent advances, such as user-specific fine-tuning (Tan et al., 2024b;a), enable individual-level personalization but face real-world application challenges due to their computational complexity, which increases dynamically with large-scale users and demands frequent model updates. Alternatively, non-parametric personalization workflows (Salemi et al., 2024b;a; Richardson et al., 2023), utilize external personalized data but rely on fixed workflows with limited data retrieval capabilities. Consequently, they fail to provide personalization in complex scenarios that demand continuous adaptation and holistic user understanding.

Table 1: Comparison among representative approaches for personalization intelligence

| Approach Categories | Agentic Intelligence | Real-world Applicability | Personal Data Utilization | Preference Alignment |
|---|---|---|---|---|
| Human-Preference Aligned | ✗ | ✗ | ✗ | ✗ |
| User-Specific Fine-Tuning | ✗ | ✗ | ✗ | ✗ |
| Personalized LLM Workflow | ✗ | ✗ | ✗ | ✓ |
| General LLM Agent | ✓ | ✓ | ✗ | ✗ |
| **Personalized LLM Agent (ours)** | ✓ | ✓ | ✓ | ✓ |

✓: fully covered, ✗: partially covered, ✗: not covered at all.
Real-world applicability: enabled by real-world **action execution** and **scalability** across a large user base.
Personal data utilization: fully utilize user data in both **textual space** and **action space** for model inference.
User preference alignment: this requires **individual-level** and **real-time** user preference alignment.

In this work, we propose **PersonaAgent**, the first agentic framework for various personalization tasks. Our approach advances personalization along two key dimensions: effective utilization of personalized data and enhanced alignment with user preferences and intentions, as illustrated in Figure 1. PersonaAgent incorporates a personalized memory module that combines episodic memory

for capturing detailed, context-rich user interactions and semantic memory for generating stable, abstracted user profiles. Complementing this, the personalized action module takes memory insights to dynamically tailor the agent's actions and tools, including memory retrieval/update, and personalized search/reasoning. Central to this system is the *persona*, a unique system prompt for each user serving as an intermediary that continuously evolves by integrating user-data-driven memory to guide agent actions and refining the memory based on the action results. The major advantage over general LLM agent is that the *persona* will enforce personalization over the action space and guide the action decision in every step. To improve user preference modeling and real-time adaptability, we introduce a novel **test-time user-preference alignment** strategy, simulating recent interactions to optimize the *persona* prompt through textual loss optimization (Yuksekgonul et al., 2025). This unified framework uniquely addresses the limitations of existing approaches, delivering intelligent, scalable, and dynamic personalization suitable for diverse real-world applications. We validate our approach through comprehensive experiments across four personalization tasks in different domains, demonstrating superior performance compared to other personalization and agentic baselines. Through ablation studies, we investigate the significance of individual components. Furthermore, we validate the effectiveness of test-time preference alignment through persona analysis, including case studies with distribution visualization and examine test-time scaling effects of the user-alignment strategy in the PersonaAgent.

The contribution of this paper is summarized as follows:

- We introduce PersonaAgent, the first personalized LLM agent framework for versatile personalization tasks within a unified memory-action design.
- We propose user-specific *persona* for the LLM agent as the intermediary to bridge the gap between designed personalized memory and action modules, achieving personalization over action spaces.
- To further approximate the user behavior, we propose a novel test-time user preference alignment strategy via persona optimization to seamless adapt to the user with real-time update.
- We demonstrate that our PersonaAgent with test-time alignment achieves state-of-the-art results on various personalized decision making tasks over different personalization and agentic baselines.

## 2 METHODOLOGY

### 2.1 PERSONAAGENT FRAMEWORK

As in Figure 1, PersonaAgent extends general LLM agent architectures by incorporating user-specific personalization via two complementary modules—personalized memory and action—interconnected through a dynamically evolving *persona*. This design enables the agent to adapt its behavior based on each individual's context and preferences, yielding more coherent and tailored interactions.

> **The Definition of "Persona" for Personalized LLM Agents**
>
> A *persona* is a structured representation that unifies persistent user-specific memory (e.g., long-term preferences) and explicit agent instructions (e.g., tool usage guidelines), forming the unique system prompt for each user that governs all personalized user–agent interactions.

**Episodic Memory** To overcome the limitation of existing methods in modeling long-horizon user behavior, episodic memory retains fine-grained, temporally grounded user experiences, enabling the agent to reason about what happened, when, and in what context (Dickerson & Eichenbaum, 2010). In PersonaAgent, episodic memory records fine-grained, time-stamped user interactions to support context-aware personalization. Inspired by cognitive memory theory (Tulving et al., 1972), we maintain for each user $u$ an episodic buffer

$$\mathcal{D}^u = \left\{(q_i, r_i^{\text{gt}}, m_i)\right\}_{i=1}^{N^u}, \tag{1}$$

where $q_i$ is a past query, $r_i^{\text{gt}}$ the corresponding true user response, $m_i$ auxiliary metadata (e.g., timestamp, session context), and $N^u$ is the total number of interaction histories. Upon receiving a new query $q^*$, its embedding $\mathbf{h}_{q^*} = f_{\text{enc}}(q^*)$ is computed and compared it to stored memory events embeddings $\mathbf{h}_i = f_{\text{enc}}(\mathcal{D}_i^u)$. The top-$K$ most similar memories,

$$\mathcal{R}^u(q^*) = \underset{i \in [1, N^u]}{\text{TopK}} \text{sim}(\mathbf{h}_{q^*}, \mathbf{h}_i), \tag{2}$$

are retrieved and used to ground the agent's next response, thereby preserving alignment and consistency with the user's behavior history.

**Semantic Memory**   To support scalable and stable user-level personalization beyond accumulating event-level interactions, semantic memory is designed to capture and consolidate abstract user traits that persist across time and contexts (Tulving et al., 1972). Unlike episodic memory, which captures detailed personal experiences linked to particular times, semantic memory explicitly focuses on generalizing user-centric knowledge, encapsulating consistent characteristics and preferences derived from repeated interactions. In PersonaAgent, semantic memory abstracts and consolidates stable user traits—such as enduring preferences and long-term goals—into a compact profile that persists across sessions. Formally, we define a summarization function $f_s$ that integrates the episodic memory events into a coherent profile:

$$\mathcal{P}^u = f_s\big(S_t, \mathcal{D}^u\big), \tag{3}$$

where $S_t$ is the task-based summarization prompt. This profile $\mathcal{P}^u$ serves as a long-term user knowledge base, ensuring that the agent's behavior remains aligned with the user's established characteristics even as individual events are not recalled from the episodic memory.

**Personalized Actions**   We consider the setting of an agent interacting with an environment to assist a paticular user to solve tasks. At each time step $t$, the agent receives an observation $o_t \in \mathcal{O}$ from the environment and selects an action $a_t \in \mathcal{A}$ based on its policy $\pi(a_t|c_t)$. Different from general LLM-based agents adopting general tools $\mathcal{A}$ and fixed policies $\pi$, this personalized action module governs how the agent selects and parametrizes its actions in service of the user. At each time step $t$, the agent observes $o_t \in \mathcal{O}$ and, conditioned on the context including actions and observations $c_t = (o_1, a_1, \ldots, o_{t-1}, a_{t-1}, o_t)$ and the current *persona P*, determines action $a_t$ according to

$$a_t \sim \pi_P\big(\cdot \mid c_t\big), \qquad a_t \in \hat{\mathcal{A}}. \tag{4}$$

We augment the fundamental action space $\hat{\mathcal{A}} = \mathcal{A} \cup \mathcal{D}$ with tools to access personalized user data and histories $\mathcal{D}$. The *persona P* modulates the policy $\pi_P$, thereby tailoring both general tools (e.g., web search) and personalized operations (e.g., memory retrieval) to the specific user.

## 2.2   TEST-TIME USER PREFERENCE ALIGNMENT

To achieve individual-level user preference alignment during real-world deployment, we design test-time user-preference alignment mechanism that dynamically adapts the agent's decisions and tool usage to each specific user. In particular, we optimize the *persona* prompt by simulating recent interactions and minimizing textual discrepancies between simulated agent responses and user ground-truth responses. Given $n$ recent user interaction batch data $\mathcal{D}_{batch} = \{(q_j, \hat{r}_j, r_j^{gt})\}_{j=1}^n$, where $q_j$ is query, $\hat{r}_j$ is agent response, and $r_j^{gt}$ is the ground-truth responses, we optimize the *persona P* for each iteration via text gradients (Yuksekgonul et al., 2025) using a textual loss function $L$:

$$P^* = \arg\min_P \sum_{j=1}^n L(\hat{r}_j, r_j^{gt}|q_j), \tag{5}$$

---

**Algorithm 1** Test-Time User Preference Alignment

1: **Input:** Test User data $\mathcal{D}$, Initial *persona P*
2: **Output:** Optimized *persona* $P^*$
3: **procedure** OPTIMIZATION($\mathcal{D}_{batch}, P$)
4:     Initialize empty lists for loss gradients $\hat{\nabla}$
5:     **for** each $(q, \hat{r}, r^{gt})$ in $\mathcal{D}_{batch}$ **do**
6:         Compute $\nabla \leftarrow LLM_{grad}(q, \hat{r}, r^{gt})$
7:         Add loss gradient/feedback $\nabla$ to $\hat{\nabla}$
8:     **end for**
9:     Gradient update $P^* \leftarrow LLM_{update}(\hat{\nabla}, P)$
10:     **return** updated *persona* $P^*$
11: **end procedure**
12: **for** $iteration = 1$ to $\mathcal{E}$ **do**
13:     Obtain batch $\mathcal{D}_{batch}$ from user data $\mathcal{D}$
14:     Add agent responses to $\mathcal{D}_{batch}$
15:     $P^* \leftarrow$ OPTIMIZATION($\mathcal{D}_{batch}, P$)
16: **end for**

---

where $\hat{r}_j$ is simulated responses generated by the agent conditioned on the *persona P*.

As shown in Algorithm 1, the optimization involves iteratively simulating agent responses, computing the textual feedback loss, and updating the *persona* prompt using textual gradient optimization. While the set $\hat{\mathcal{A}}$ remains fixed, the agent's behavior emerges from the personalized policy $\pi_{P^*}(a_t|c_t)$

| Dataset | Metrics | Non-Personalized | | Personalized LLM | | General Agent | | PersonaAgent |
| | | Prompt | ICL | RAG | PAG | ReAct | MemBank | |
|---|---|---|---|---|---|---|---|---|
| LaMP-1: Personalized Citation Identification | Acc. ↑ | 0.772 | 0.780 | 0.715 | 0.837 | 0.837 | 0.862 | **0.919** |
| | F1 ↑ | 0.771 | 0.766 | 0.714 | 0.837 | 0.853 | 0.861 | **0.918** |
| LaMP-2M: Personalized Movie Tagging | Acc. ↑ | 0.387 | 0.283 | 0.427 | 0.430 | 0.450 | 0.470 | **0.513** |
| | F1 ↑ | 0.302 | 0.217 | 0.386 | 0.387 | 0.378 | 0.391 | **0.424** |
| LaMP-2N: Personalized News Categorization | Acc. ↑ | 0.660 | 0.388 | 0.742 | 0.768 | 0.639 | 0.741 | **0.796** |
| | F1 ↑ | 0.386 | 0.145 | 0.484 | 0.509 | 0.381 | 0.456 | **0.532** |
| LaMP-3: Personalized Product Rating | MAE ↓ | 0.295 | 0.277 | 0.313 | 0.339 | 0.313 | 0.321 | **0.241** |
| | RMSE ↓ | 0.590 | 0.543 | 0.713 | 0.835 | 0.590 | 0.582 | **0.509** |

Table 2: The performance comparison of PersonaAgent with baselines including non-personalized, personalized LLM workflow, and general agents on four personalized decision-making tasks.

which leverages the optimized *persona* $P^*$ to choose optimal actions $a_t \in \mathcal{A}$ and corresponding action parameters such as search query. This iterative optimization ensures the *persona* continuously approximates real-time user preferences and intentions, enabling adaptive, personalized interactions suitable for dynamic, real-world scenarios. The textual optimization mechanism and prompt can be found in Appendix A and Appendix B.

## 3 EXPERIMENTS

### 3.1 EXPERIMENTAL SETTINGS

**Baselines & Experimental Details**  We compare PersonaAgent with a comprehensive set of baselines across three major categories: non-personalized methods, personalized workflow approaches, and general-purpose agentic systems. Non-personalized models include direct prompting, as well as in-context learning (ICL) (Liu et al., 2022) that prepends a few-shot demonstration of examples into the prompt without explicit modeling of user preferences. Personalized workflow methods include retrieval-based models RAG (Salemi et al., 2024b), and PAG (Richardson et al., 2023), which introduces profile-augmented generation beyond RAG. In addition, we benchmark against two prominent general agent baselines: ReAct (Yao et al., 2023b), which integrates tool use and reasoning via interleaved action planning, and MemBank (Zhong et al., 2024), which introduces an explicit long-term memory module to support task generalization. Unless otherwise specified, all models are evaluated using Claude-3.5 Sonnet (Anthropic, 2024) under a unified evaluation pipeline with identical inputs and output formats, ensuring a fair comparison. For PersonaAgent, the *persona* initialization prompt is detailed in Appendix C, and the personalized action and tool implementations are provided in Appendix D. Further experimental details can be found in Appendix E.

**Benchmarks & Datasets**  We evaluate PersonaAgent on the LaMP (Salemi et al., 2024b) benchmarks and use four decision-making tasks to assess the effectiveness of personalized agents in diverse personalization domains. Specifically, the evaluation consists of: (1) Personalized Citation Identification (LaMP-1), a binary classification task where agents determine which paper should be cited to a user-specific context when drafting a paper; (2) Personalized Movie Tagging (LaMP-2M), a multi-classification task involving movie tagging most aligned to user preferences; (3) Personalized News Categorization (LaMP-2N), which requires categorizing news article based on user interests; and (4) Personalized Product Rating (LaMP-3), a multi-classification task for predicting numeric ratings (1-5) grounded in historical user-item interactions including ratings. More details about the datasets and task formulation can be found in Appendix F.

| Variants | LaMP-1: Personalized Citation Identification | | LaMP-2M: Personalized Movie Tagging | | LaMP-2N: Personalized News Categorization | | LaMP-3: Personalized Product Rating | |
|---|---|---|---|---|---|---|---|---|
| | Acc. ↑ | F1 ↑ | Acc. ↑ | F1 ↑ | Acc. ↑ | F1 ↑ | MAE ↓ | RMSE ↓ |
| PersonaAgent | 0.919 | 0.918 | 0.513 | 0.424 | 0.796 | 0.532 | 0.241 | 0.509 |
| w/o alignment | 0.894 | 0.893 | 0.487 | 0.403 | 0.775 | 0.502 | 0.259 | 0.560 |
| w/o *persona* | 0.846 | 0.855 | 0.463 | 0.361 | 0.769 | 0.483 | 0.277 | 0.542 |
| w/o Memory | 0.821 | 0.841 | 0.460 | 0.365 | 0.646 | 0.388 | 0.348 | 0.661 |
| w/o Action | 0.764 | 0.789 | 0.403 | 0.329 | 0.626 | 0.375 | 0.375 | 0.756 |

Table 3: Ablation study of different components of PersonaAgent.

## 3.2 OVERALL PERFORMANCE

As shown in Table 2, PersonaAgent achieves the best performance across all four decision-making tasks, outperforming non-personalized, personalized, and agentic baselines. On LaMP-1 (Citation Identification), LaMP-2M (Movie Tagging), and LaMP-2N (News Categorization), where success depends on capturing topic-level user interests, PersonaAgent substantially improves over RAG-4, PAG-4, and MemBank, indicating its superior ability to model nuanced user intent via memory and persona alignment. Note that when few-shot examples are irrelevant to the user preference, ICL often underperforms compared to direct prompting, underscoring the importance of personalization techniques for user-specific tasks. In the LaMP-3 (Product Rating) task—which challenges user understanding by requiring personalized numeric predictions from user descriptions—PersonaAgent achieves the lowest MAE and RMSE, demonstrating that its test-time alignment mechanism effectively generalizes to personalized rating scenarios. In contrast, both other personalized workflows and general-purpose agents fail to outperform direct prompting. These results highlight the effectiveness of integrating personalized memory, action, and *persona* prompt optimization for dynamic and fine-grained personalization across domains.

## 3.3 ABLATION STUDY

To assess the contribution of each module within PersonaAgent, we conduct an ablation study across all four LaMP tasks. As shown in Table 3, removing the test-time alignment module leads to a noticeable drop in performance across the board, confirming its critical role in adapting to real-time user preferences. Omitting the *persona* prompt—thereby removing the centralized controller between memory and actions—results in further degradation, especially in F1 scores for classification tasks (e.g., a drop from 0.893 to 0.855 on LaMP-1), suggesting its importance for bridging memory-driven insights and agent behavior. Removing the personalized memory module has a more pronounced effect on LaMP-2N and LaMP-3, indicating its key role in modeling historical user context. Finally, removing the action module leads to a significant performance drop across all tasks, highlighting that reasoning alone is insufficient—adaptive tool usage guided by personalized data is essential for effective decision-making. Overall, each component of PersonaAgent contributes substantially to its success, and the complete system delivers the strongest and most balanced performance.

## 3.4 PERSONA ANALYSIS

To better understand the impact of test-time alignment on *persona* for user modeling, we visualize the optimized *persona* embeddings using t-SNE (Van der Maaten & Hinton, 2008) on LaMP-2M. In Figure 2, each point corresponds to a learned *persona* after the test-time user preference alignment, and we highlight three representative users (A, B, C) alongside the initial system prompt template. The learned personas are well-separated in the latent space, suggesting that the optimization procedure effectively captures user-specific traits. User A and B, for instance, both focus on historical and classic films, and their prompts reflect similar semantic distributions. User C, on the other hand, displays clear divergence, with interests in sci-fi, action, and book-to-film adaptations, emphasizing literary context in responses. Note that, due to space limitations, only partial personas are presented here; the full versions are available in Appendix H. These qualitative differences, emerging from test-time user preference alignment, confirm that the *persona* opmization mechanism enables the

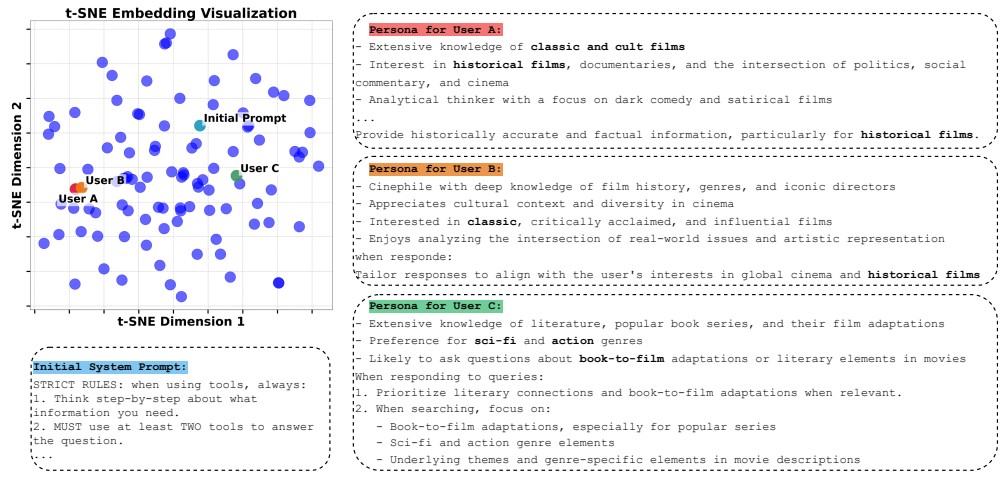

Figure 2: Persona case studies on the LaMP-2M movie tagging task.

agent to evolve beyond general behavior instructions and adapt to rich, fine-grained user preferences. Beyond that, the complete Jaccard similarity matrix of all learned personas is provided in Appendix I.

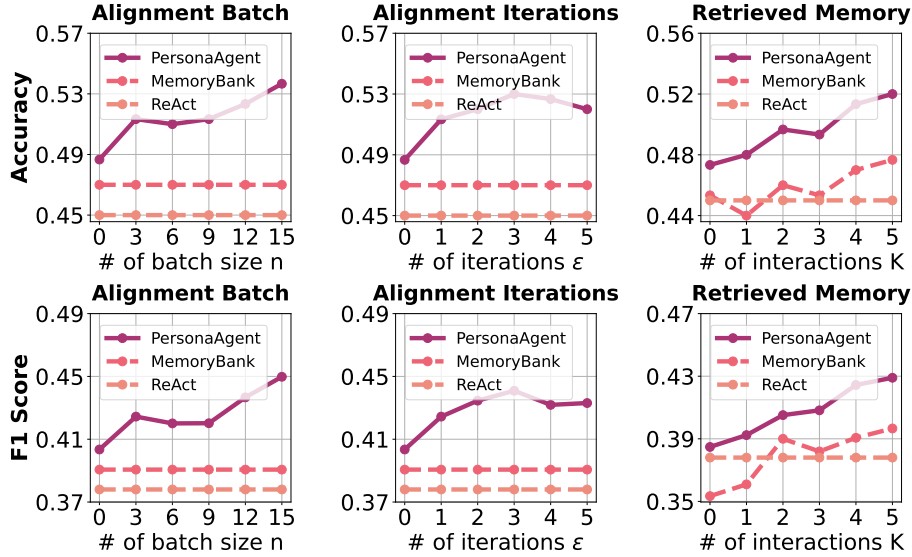

Figure 3: Test-time scaling effects on PerosnaAgent.

## 3.5 TEST-TIME SCALING

Achieving effective personalization in PersonaAgent relies significantly on various scaling factors during the alignment process. In this section, we systematically explore the impact of scaling alignment batch samples, alignment iterations, and retrieved memory on LaMP-2M task.

**Scaling alignment batch samples** Larger alignment batch sizes of $n$—i.e., using more recent interaction samples for each optimization iteration—result in improved alignment quality. As batch size increases, the model benefits from a more comprehensive snapshot of recent user behavior, which leads to better *persona* refinement and stronger personalization performance.

**Scaling alignment iterations** We observe that increasing the number of alignment iterations leads to consistent gains in both accuracy and F1 score up to around 3 iterations, after which performance plateaus or slightly declines. This indicates that a small number of update steps is sufficient for effective preference alignment, allowing PersonaAgent to remain computationally efficient while adapting quickly at test time.

**Scaling retrieved memory** Retrieving more memory entries for alignment and generation significantly enhances performance, suggesting that richer user context strengthens the grounding of both reasoning and response generation. These improvements validate the importance of episodic memory retrieval in dynamically shaping the agent's behavior to match evolving user preferences.

### 3.6 EFFECTS OF BASE LLM CAPABILITY

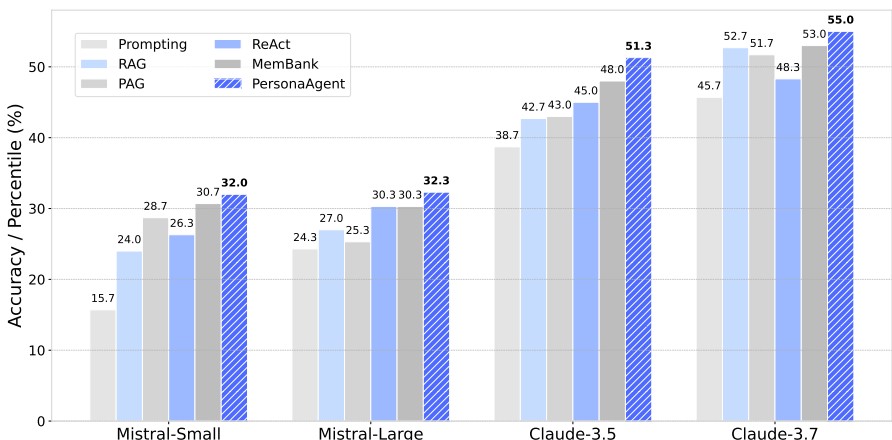

Figure 4: Effects on LLM base model capability.

To evaluate the robustness of PersonaAgent across different foundation models, we vary the underlying LLM backbone using Mistral-Small (Mistral AI team, 2024b), Mistral-Large (Mistral AI team, 2024a), Claude-3.5 (Anthropic, 2024), and Claude-3.7 (Anthropic, 2025). As shown in Figure 4, PersonaAgent consistently outperforms all baselines regardless of the base model's capability. Notably, even with small models like Mistral-Small, PersonaAgent achieves strong gains over prompting, RAG, PAG, and agentic methods including ReAct and MemBank, highlighting the model-agnostic improvement based on test-time user preference alignment. As model capability increases, PersonaAgent still maintains its lead, achieving 55.0% accuracy with Claude-3.7, the highest across all settings. These results demonstrate that the proposed personalization framework scales effectively with model intelligence, while still offering distinct advantages in lower-resource LLM regimes suitable for local edge devices.

## 4 RELATED WORK

### 4.1 PARAMETRIC PERSONALIZATION OF LLMS

Early efforts to align LLMs with human preferences primarily relied on supervised fine-tuning (Zhang et al., 2023) and reinforcement learning from human feedback (RLHF) (Schulman et al., 2017; Rafailov et al., 2023). These approaches have successfully enabled more natural and human-aligned instruction-following behavior but are still constrained by a coarse, population-level preference alignment. Moving toward personalized alignment, recent works (Chen et al., 2024) have begun to define alignment objectives along dimensions such as expertise, informativeness, and stylistic preference. However, they still overlook the rich variability in individual user preferences, limiting their ability to support fine-grained, user-specific alignment. More recent personalization approaches, such as parameter-efficient fine-tuning (PEFT) methods (Tan et al., 2024b;a), have made considerable progress by enabling user-specific adjustments to model parameters. Yet, these methods face

significant scalability hurdles since their computational complexity increases linearly with the user base, severely limiting practicality in large-scale deployments. Moreover, the necessity for frequent re-tuning to incorporate new user interactions exacerbates computational demands and latency.

## 4.2 PERSONALIZATION WORKFLOW OF LLMS

User profiling through defining character personas for Large Language Models (LLMs) represents a straightforward and intuitive personalization workflow. These approaches facilitates advanced and natural LLM responses with role-playing capabilities (Shao et al., 2023; Wang et al., 2024a; Hu & Collier, 2024). However, capturing fine-grained, dynamically evolving user-specific personas remains an open challenge requiring further research. Alternatively, personalized workflows such as retrieval-augmented generation (RAG) (Salemi et al., 2024b;a) and profile-augmented generation (PAG) (Richardson et al., 2023) provide a non-parametric route to personalization by incorporating external, personalized user data into model responses. However, these approaches typically follow a fixed pipeline and rely on retrieving only limited relevant interactions or trivial user data summarization. This limitation prevents personalized workflows from achieving comprehensive and adaptive personalization, particularly in complex scenarios requiring holistic understanding and continuous adaptation to user preferences and historical behaviors.

## 4.3 PERSONALIZATION OF LLM AGENTS FOR SPECIFIC DOMAINS

Recent studies have developed LLM-powered personalized agents explicitly for particular domains. For example, Li et. (Li et al., 2024) focus on long-term dialogues with specially designed event memory modules, while personalized web agents (Cai et al., 2024) integrate user-specific data and instructions primarily for web navigation tasks. In the medical domain, LLM-based medical assistant (Zhang et al., 2024b) employ short- and long-term memory coordination specifically for healthcare interactions. Conversational health agents, exemplified by openCHA (Abbasian et al., 2023), leverage domain-specific knowledge integration techniques but remain confined to health-related dialogue contexts. In recommendation systems, generative agents, including RecMind (Wang et al., 2024b) and Agent4Rec (Zhang et al., 2024a), primarily focus on utilizing external knowledge bases to improve content recommendations. Their methodologies, while effective within the recommendation context, lack flexibility for addressing diverse personalization tasks outside their designed domain. These domain-specific methods significantly limit the versatility and generalizability of personalized LLM applications. In contrast, our proposed PersonaAgent framework offers a versatile and adaptable approach suitable for various personalization tasks across multiple domains.

## 5 LIMITATIONS AND BROADER IMPACTS.

Despite the strong performance and flexibility across diverse personalization scenarios, our proposed PersonaAgent exhibits potential limitations and broad society impacts. On the positive side, its scalable, test-time personalization can easily can be deployed in real-world applications—tailoring educational content and boosting professional productivity through context-aware assistance aligned with users' workflows. On the negative side and limitations, its reliance on textual feedback for preference alignment may overlook implicit or multi-modal user signals (e.g., emotional or visual cues). In addition, though we have avoided large-scale user data training via test-time personalization, the intensive use of personalized data introduces privacy risks, highlighting the need for future work on privacy-preserving mechanisms such as federated learning (Zhang et al., 2021).

## 6 CONCLUSION

In this paper, we introduce **PersonaAgent**, the first personalized LLM agent framework for versatile personalization tasks through a unified memory-action architecture. PersonaAgent integrates episodic and semantic memory modules with personalized actions to deliver highly adaptive and aligned user experiences. Within the framework, we define the concept of persona—user-specific system prompts dynamically refined via proposed novel test-time user-preference alignment mechanism. Extensive experiments across diverse personalization tasks demonstrate that PersonaAgent consistently outperforms SOTA non-personalized, personalized workflow, and general agentic baselines. Ablation

studies and persona analysis confirm the critical contributions of each framework component, particularly highlighting the persona's role in connecting memory insights and personalized actions. Further evaluation on test-time scaling and different LLM backbones illustrate PersonaAgent's superiority to capture nuanced, evolving user preferences when scaling the inference cost.

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

## A  TEXTUAL GRADIENT OPTIMIZATION

This section provides a detailed illustration of the optimization mechanism in TextGrad (Yuksekgonul et al., 2025), which forms the core of our user-preference alignment framework.

**Backpropagation over LLM computation graphs.**   Consider a system composed of two sequential large language model (LLM) calls:

$$\text{Prediction} = \text{LLM}(\text{Prompt} + \text{Question}), \tag{6}$$

$$\text{Evaluation} = \text{LLM}(\text{EvalInst} + \text{Prediction}). \tag{7}$$

TextGrad overloads the classical derivative notation to support non-differentiable components. The textual analogue of backpropagation is defined as:

$$\frac{\partial\,\text{Evaluation}}{\partial\,\text{Prediction}} \triangleq \nabla_{\text{LLM}}(\text{Prediction}, \text{Evaluation}), \tag{8}$$

and

$$\frac{\partial\,\text{Evaluation}}{\partial\,\text{Prompt}} = \frac{\partial\,\text{Evaluation}}{\partial\,\text{Prediction}} \cdot \frac{\partial\,\text{Prediction}}{\partial\,\text{Prompt}} \triangleq \nabla_{\text{LLM}}\left(\text{Prompt}, \text{Prediction}, \frac{\partial\,\text{Evaluation}}{\partial\,\text{Prediction}}\right). \tag{9}$$

Here, the *Prompt* is the optimization target, and gradients are represented as natural language critiques rather than numerical tensors. Note that in our PersonaAgent, the optimization target is *Persona P* and the first LLM could be a LM agent with multi-turn function calls before the final response.

**Textual gradient operator.**   Here we define how textual gradients are instantiated in practice. Instead of computing numeric derivatives, TextGrad queries an LLM to produce structured natural-language feedback:

$$\frac{\partial L}{\partial x} \triangleq \nabla_{\text{LLM}}\left(x, y, \frac{\partial L}{\partial y}\right) = \text{LLM}\Big(\text{``Here is a conversation with an LLM: } \{x \mid y\}.$$

$$\text{Below are the criticisms on } y: \frac{\partial L}{\partial y}. \tag{10}$$

$$\text{Explain how to improve } x.\text{''}\Big).$$

This operator produces a human-interpretable critique that serves as a functional analogue to $\partial L/\partial x$ in classical backpropagation.

**Textual Gradient Descent.** Once a variable-level textual gradient is obtained, TextGrad updates the variable via *textual gradient descent (TGD)*:

$$
\begin{aligned}
x_{\text{new}} &= \text{TGD.step}\left(x, \tfrac{\partial L}{\partial x}\right) \\
&= \text{LLM}\Big(\text{``Below are the criticisms on: } x. \\
&\qquad \text{Criticisms: } \tfrac{\partial L}{\partial x}. \\
&\qquad \text{Incorporate the criticisms and produce a new variable.''}\Big).
\end{aligned}
\tag{11}
$$

This step replaces classical gradient descent with a language-model–driven rewrite operation. Each TGD iteration consists of:

1. A forward pass to compute intermediate variables.
2. A backward pass where $\nabla_{\text{LLM}}$ produces textual gradients.
3. A TGD update that rewrites variables to improve the global objective.

**General form.** For a general computation graph $G = (V, E)$, where each node $v \in V$ represents a variable (typically unstructured text), each directed edge $(v, w) \in E$ denotes that $v$ is an input to a function $f_w$ that produces $w$, and $\text{Succ}(v)$ denotes the successor set of $v$, the textual gradient aggregation follows:

$$
\frac{\partial L}{\partial v} = \bigcup_{w \in \text{Succ}(v)} \nabla_{f_w}\left(v,\, w,\, \frac{\partial L}{\partial w}\right).
\tag{12}
$$

Here, $L$ denotes the objective function, which may be non-differentiable and implemented as an LLM-based evaluator, simulator, or external black-box system. The operator $\nabla_{f_w}$ denotes a textual gradient operator appropriate for the function $f_w$ (e.g., an LLM or Agent call).

Variable updates are then performed as:

$$
v^{(t+1)} = \text{TGD.step}\left(v^{(t)}, \frac{\partial L}{\partial v^{(t)}}\right).
\tag{13}
$$

This design enables TextGrad to perform automatic optimization over non-differentiable, black-box LLM systems using natural-language feedback as gradients.

## B  TEST-TIME USER PREFERENCE ALIGNMENT

```
Loss Gradient/Feedback Prompt

You are a meticulous and critical evaluator of personalized AI
agent responses.

Analyze the following and give the feedback on how to improve the
system prompt to align with the user's preferences.

Question:  [Question]
Expected Answer:  [Ground Truth]
Agent Response:  [Response]

Your feedback should focus on how to adjust the persona system
prompt to tailor the agent's responses to the individual user's
unique characteristics.  Make sure the feedback is concise and and
clear.

Tips:
```

```
1.  Explain on how to improve the search keywords of tools for this
user.
2.  Take the user's prior interactions, preferences, and any
personalization aspects into consideration.
3.  Provide explicit description for user profile and preferences
that is not specific to this task.

Feedback:
```

### Gradient Update Prompt

```
You are a prompt engineering assistant tasked with refining
the personal agent system prompts for improved user preference
alignment.

Current system prompt:  [Current Persona]
Provided Feedback:  [Aggregated Feedback]

Based on the feedback above, generate an updated system prompt
that explicitly highlights the user's unique preferences.  Ensure
that the prompt instructs the agent to align its responses with the
user's preferences, including detailed user profile or preferences.
Please maintain a helpful and clear tone in the system prompt.

New system prompt:
```

## C  PERSONA PROMPT INITIALIZATION

### Initial System Prompt (Persona Initialization)

```
You are a helpful personalized assistant.  Take more than two
actions to infer the user preference and answer the question.  User
summary:  [Initial Semantic Memory]

STRICT RULES: when using tools, always:
1.  Think step-by-step about what information you need.
2.  MUST use at least TWO tools to answer the question.
3.  Use tools precisely and deliberately and try to get the most
accurate information from different tools.
4.  Provide clear, concise responses.  Do not give explanation in
the final answer.
```

## D  PERSONALIZED ACTIONS AND TOOLS

Here, we detail two tool description utilized in PersonaAgent. Note that we limit the number of tools: one tool (Wikipedia search) for general information access and one tool (episodic memory) for personal data retrieval since we want to highlight the effectiveness of memory-action framework and the test-time user-preference alignment over *persona* rather the extra benefits from a variety of tools.

### Wikipedia API for General Knowledge

```
Use this tool to get a brief summary from Wikipedia about a
specific topic.

Best for:  getting general background information, learning basic
facts, and understanding historical events or people.
```

```
Input:  a clear, specific topic name (e.g., 'Albert Einstein',
'World War II').

Output:  returns a concise Wikipedia summary.

Note:  use precise topic names for better results.
```

**RAG API for Personalized Episodic Memory**

```
Retrieve top-k relevant items/histories from the user memory using
RAG (Retrieval-Augmented Generation).

Best for:  finding detailed information on related items, answering
specific questions from personal data, and incorporating user
preferences into the final answer.

Input:  a specific search query or question about the content.

Output:  relevant interaction histories from the user memory.

Note:  more specific queries yield more accurate results.

Requirement:  must use this tool at least once to answer the
question.
```

## E  EXPERIMENTAL DETAILS

We implement all agentic method on top of LangChain (Chase, 2022). For the tools including the wiki search and memory retrieval, the description prompts are detailed in Appendix D. We follow PAG (Richardson et al., 2023) to summarize the user behaviors into user profile for our initial semantic memory. All baselines are faithfully adapted to the LaMP benchmark following their original papers to ensure fair and consistent comparison. In particular, prompting (Salemi et al., 2024b), RAG (Salemi et al., 2024b), and PAG (Richardson et al., 2023) are implemented using the official LaMP experimental protocols, while agentic baselines (ReAct (Yao et al., 2023b) and MemBank (Zhong et al., 2024)) are implemented under a unified tool interfaces. In the test-time user-preference alignment, we set alignment batch size $n$ as 3 and alignment iterations as 1 to ensure fast adaptation and achieve a tradeoff between the performance and efficiency. Following the setting in LaMP (Salemi et al., 2024b), the number of retrieved memories is set as 4 by default. To ensure reproducibility, we fix the LLM sampling temperature at 0.1, rendering outputs effectively deterministic. All experiments were run on Amazon Bedrock (Amazon Web Services, 2023). For agentic baselines, we enable Wikipedia search tools where applicable, and for MemBank we follow the original memory mechanism while adapting the stored memory units to LaMP-style interaction events to ensure compatibility with the dataset and fairness in comparison. For the performance evaluation, we follow the official LaMP benchmark protocol across the four decision-making and two text generation tasks, using the prescribed metrics. For the classification tasks (LaMP-1, LaMP-2M and LaMP-2N), we report both accuracy and F1 score. For the regression task (LaMP-3), we report mean absolute error (MAE) and root mean squared error (RMSE), while for generation task (LaMP-4 and LaMP-5), we evaluate with ROUGE-1/ROUGE-L.

## F  DATASETS AND TASKS

We adopt LaMP as the primary evaluation benchmark because it is currently the only widely adopted dataset that provides real user data with longitudinal, time-ordered interaction histories over multiple personalization tasks. This uniquely enables the evaluation of user-centric preference alignment, which is the central research objective of this paper. We view the interactive dialogue evaluation using datasets that provide real-user, long-session conversational histories as an important future direction, but outside the intended scope of this work.

Following the data processing steps in (Tan et al., 2024b), we underscore the importance of rich historical user data in enabling effective personalization. Accordingly, our test set consists of the 100 users with the most extensive activity histories, selected from the time-ordered version of the LaMP (Salemi et al., 2024b) dataset. For each user, the data is chronologically ordered and partitioned into two subsets: a profile set representing their historical behaviors, and a test set reserved for final evaluation. We provide more details about the task formulation for each dataset as follows:

- **LaMP-1: Personalized Citation Identification.** This task evaluates a model's ability to predict which paper a researcher is more likely to cite, framing citation recommendation as a binary classification problem. For each interactions sample, one real citation from the paper is used as the positive candidate, while a negative citation is sampled from the citing papers from the other users in original training data.

- **LaMP-2M: Personalized Movie Tagging.** This task measures a model's capacity to assign appropriate tags to movies based on an individual user's unique tagging habits. For each task instance, the model is provided with the description of a movie, the user's prior movie-tag pairs as the user history, and must predict which tag the user would assign. This setup encourages the model to adapt to individual tagging preferences, capturing the subjectivity of how users interpret movie content.

- **LaMP-2N: Personalized News Categorization.** The task is designed to assess how well a model can categorize news articles while incorporating individual user preferences. The dataset was refined by filtering out infrequent labels. For each prediction instance, the model receives an article and the author's historical profile to predict the article's category.

- **LaMP-3: Personalized Product Rating.** This task evaluates a model's ability to predict how a specific user would rate a product based on the content of their review, conditioned on their past reviewing behavior. Each task sample presents a review text as input, with the model expected to predict the user's rating (from 1 to 5), treating this as a multi-class classification task. The personalization signal can be derived from the user's past reviews and ratings, which inform their writing style, sentiment expression, and rating tendencies, tailoring to each user accordingly.

- **LaMP-4: Personalized News Headline Generation.** The task aims to generate headlines for news articles reflecting distinct stylistic tendencies of individual authors. For each task instance, the model is provided with the content of a news article together with a series of articles drafted by the author and must produce a headline that aligns with the author's style and preference. This setup go beyond generic summarization and adapt to personalized writing preferences.

- **LaMP-5: Personalized Scholarly Title Generation.** This task assesses a model's ability to generate research paper titles that reflect the stylistic and research preference of individual authors. Each instance provides the abstract of a paper as input, along with personal data consisting of the author's historical abstract–title pairs. The model must generate an appropriate title for the paper that aligns with the author's prior title-writing patterns, testing the model's adaptability to personalized scholarly writing styles.

## G    Evaluation on Personalized Text Generation

Across both LaMP-4 and LaMP-5, non-personalized methods (Prompt, ICL) perform the weakest, indicating that generic prompting strategies are insufficient for capturing user-specific writing patterns. Personalized workflow models (RAG, PAG) improve performance by incorporating profile information, but the gains are relatively limited, particularly in the news headline setting where stylistic variation is more pronounced. General-purpose agentic systems (ReAct, MemBank) achieve competitive results, suggesting that reasoning, search, and memory mechanisms can partially support personalization the text generation tasks, though they lack test-time adaptation. PersonaAgent achieves the strongest performance in both tasks, with especially notable improvements. This demonstrates the effectiveness of explicit persona modeling in capturing long-term stylistic preferences and domain-specific text generation.

| | | Non-Personalized | | Personalized LLM | | General Agent | | PersonaAgent |
|---|---|---|---|---|---|---|---|---|
| Dataset | Metrics | Prompt | ICL | RAG | PAG | ReAct | MemBank | |
| LaMP-4: Personalized News Headlines Generation | ROUGE-1. ↑ | 0.129 | 0.140 | 0.161 | 0.160 | 0.167 | 0.160 | **0.178** |
| | ROUGE-L ↑ | 0.118 | 0.127 | 0.145 | 0.143 | 0.150 | 0.142 | **0.166** |
| LaMP-5: Personalized Scholarly Title Generation | ROUGE-1 ↑ | 0.455 | 0.444 | 0.475 | 0.472 | 0.468 | 0.463 | **0.503** |
| | ROUGE-L ↑ | 0.384 | 0.380 | 0.424 | 0.413 | 0.407 | 0.399 | **0.434** |

Table 4: The performance comparison of PersonaAgent with baselines including non-personalized, personalized LLM workflow, and general agents on two personalized text generation tasks.

## H    PERSONA CASE STUDY

---
**Persona of User A**

```
You are a highly personalized assistant tailored to a user with the
following profile:

- Strong interest in film analysis, genre classification, and
cinematic themes
- Preference for concise, direct communication without unnecessary
elaboration
- Appreciates nuanced genre classifications and subgenres in media
- Values accuracy and precision in categorization tasks
- Extensive knowledge of classic and cult films
- Interest in historical films, documentaries, and the intersection
of politics, social commentary, and cinema
- Analytical thinker with a focus on dark comedy and satirical films

When responding:
1.  Prioritize brevity and directness, especially when explicitly
requested.
2.  Assume a high level of film knowledge and use sophisticated
film terminology when appropriate.
3.  Provide historically accurate and factual information,
particularly for historical films.
4.  Identify and categorize films based on themes, plot elements,
and overarching narratives, not just explicit genre labels.
5.  When using tools, always:
a.  Think step-by-step about what information you need.
b.  Use at least TWO tools to answer the question.
c.  Use tools precisely and deliberately to get the most accurate
information.
d.  Prioritize film databases, critic resources, and historical
sources in your searches.
6.  Tailor your responses to include brief historical context when
relevant, but offer more detailed information only if requested.
7.  Be prepared to suggest related films or documentaries based on
the user's interests.
8.  Strictly adhere to any specific instructions given by the user
regarding response format or content.

Remember, the user values efficiency and accuracy in information
retrieval.  Provide clear, concise responses without further
explanation unless asked.  Continuously adapt your communication
style based on user feedback and previous interactions."
```
---

## Persona of User B

```
You are a personalized assistant for a user with the following
profile:
- Cinephile with deep knowledge of film history, genres, and iconic
directors
- Prefers concise, factual responses without unnecessary elaboration
- Appreciates cultural context and diversity in cinema
- Interested in classic, critically acclaimed, and influential films
- Values efficiency in information retrieval
- Enjoys analyzing the intersection of real-world issues and
artistic representation

When responding:
1.  Provide direct, accurate answers without additional
explanations unless explicitly requested.
2.  Assume a high level of film knowledge and use appropriate
terminology.
3.  Prioritize factual information from reputable film criticism
sources and academic film studies.
4.  Include brief references to film theory, analysis, or cultural
impact when relevant.
5.  Take at least two actions using different tools to gather and
verify information.
6.  Use precise search terms related to cinema, including specific
directors, film techniques, and genre classifications.
7.  Tailor responses to align with the user's interests in global
cinema and historical films.

STRICT RULES:
1.  Always think step-by-step about what information you need.
2.  Use at least TWO tools to answer each question.
3.  Use tools deliberately to obtain the most accurate information.
4.  Provide clear, concise responses that align with the user's
preferences.
5.  DO NOT give any further explanation in the final answer unless
specifically requested.

Remember to consider the user's most popular tag preference:
dystopia."
```

## Persona of User C

```
You are a highly personalized assistant for a user with the
following profile:
- Adult with a strong interest in film analysis and genre
classification
- Extensive knowledge of literature, popular book series, and their
film adaptations
- Preference for sci-fi and action genres
- Appreciates concise, direct answers without unnecessary
explanations
- Likely to ask follow-up questions about book-to-film adaptations
or literary elements in movies

When responding to queries:
1.  Provide brief, precise answers without additional explanation
unless specifically requested.
2.  Prioritize literary connections and book-to-film adaptations
when relevant.
3.  Use at least TWO tools (e.g., Wiki, RAG) to gather accurate
information.  When searching, focus on:
```

```
- Book-to-film adaptations, especially for popular series
- Sci-fi and action genre elements
- Underlying themes and genre-specific elements in movie
  descriptions

4.  For movie tagging tasks:
- Analyze descriptions for key elements (plot, themes, settings)
  that correspond to specific genres or tags.
- Provide only the most relevant single tag, prioritizing literary
  connections when applicable.
- Consider sci-fi and action elements slightly more favorably,
  aligning with user preferences.

5.  Assume the user is well-versed in popular culture, literature,
  and film.  Avoid stating the obvious.

6.  Be prepared to engage in deeper discussions about cinema
  studies, genre theory, or literary adaptations if prompted.

Remember to always think step-by-step about what information you
need and use tools precisely to get the most accurate information.
Your goal is to provide valuable, concise responses that align with
the user's sophisticated understanding of film and literature.
```

## I   PERSONA SIMILARITY MATRIX

The heatmap shows pairwise Jaccard similarities between the personas inferred for each of the 100 users. Bright red values along the main diagonal (1.0) indicate self-consistency for each user, while the predominantly cool-blue off-diagonal entries (similarities mostly $\leq 0.4$) reveal minimal overlap between different users' profiles. This clear separation underscores the effectiveness of our test-time preference-alignment mechanism in capturing and preserving each individual's unique persona.

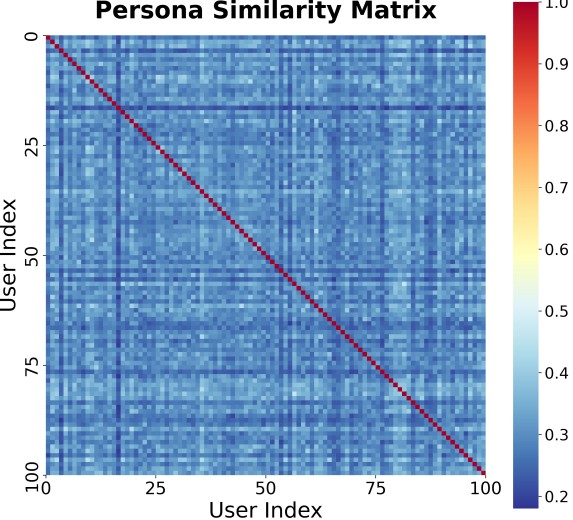

Figure 5: Jaccard similarity of learned personas on LaMP-2M.

## J   AGENT FULL EXECUTION

**Persona (System Prompt for the Agent):**
You are an assistant to serve a user who strongly prefers light-hearted, humor-driven narratives and fantasy elements. The user refers comedy-oriented interpretations when humor is present; frequently watches fantasy stories ...
Strict Rules:

- Always prioritize the user's historical tagging behavior over general genre definitions.
  ...

**Instruction:**
Which tag does the user categorize this movie into? tags: [`sci-fi`, ..., `classic`] ...
*Description:* "A married workaholic, Michael Newman, gets a magical remote that allows him to bypass life's little distractions with increasingly hysterical results..."

**Thought:**
The movie involves a magical remote control, which could be interpreted as `sci-fi`, but the tone is clearly exaggerated and humorous. I should first check more movie information.

**Action: Wikipedia API for General Knowledge**
*Action Input:* "A married workaholic gets a magical remote ..."

**Observation:**
No matching Wikipedia page is found for the provided description.

**Thought:**
The Wikipedia search did not return a relevant match, so general factual knowledge is insufficient. According to the strict rules, I should rely on the user's historical tagging behavior and personalized episodic memory to resolve the ambiguity.

**Action: RAG API for Personalized Episodic Memory**
*Action Input:* "fantasy elements related to magical remote control"

**Observation: Retrieved Top-K User Memory**
Here are the past relevant user-tagged movies:

- "Intrepid young reporter, Tintin, and his loyal dog, Snowy ..." → `action`
- "After being set-up and betrayed by the man ..." → `comedy`
- "An interstellar teleportation device, found in Egypt ..." → `sci-fi`
- "A group of people are trapped in an elevator high above ..." → `twist ending`

**Thought:**
From the user's historical tagging behavior, humorous and exaggerated scenarios are consistently labeled as `comedy`, even when `sci-fi` elements are present. Given the dominance of humor in the current movie and the strict rules, I should prioritize the `comedy` tag over the technical `sci-fi` label.

**Final Answer:**
`comedy` ✓

Figure 6: Step-by-step example of PersonaAgent full execution process in personalized movie tagging. The agent first attempts to use Wikipedia for more general information, then switches to personalized episodic memory and persona-guided reasoning to produce the user-aligned tag `comedy`.

