# OpenReview forum: "PersonaAgent: When Large Language Model Agents Meet Personalization at Test Time"
_ICLR.cc/2026/Conference — ICLR 2026 Conference Withdrawn Submission_

### Official Review · Reviewer_q71x · 2025-10-26

**Soundness:** 2
**Presentation:** 1
**Contribution:** 2
**Rating:** 4
**Confidence:** 2

**Summary:**

This paper focuses on personalized LLM agents. To address the one-size-fits-all limitations of existing approaches, it proposes PersonaAgent. The core of this framework is the persona, which acts as an intermediary between the personalized memory module (including semantic memory mechanisms) and personalized actions. Based on this framework, the paper proposes a test-time user-preference alignment strategy to optimize the persona by simulating recent interactions. Finally, the effectiveness of the proposed framework is demonstrated on the LaMP benchmark.

**Strengths:**

1. Using the persona as an intermediary between memory and action is intuitive and reasonable.
2. A complete personalized agent framework, PersonaAgent, is proposed.
3. Experimental results demonstrate the effectiveness of the proposed strategy, and detailed ablation experiments are performed.

**Weaknesses:**

1. The paper is poorly written, the method is obscure, and lacks necessary details. For example, the input to $f_{enc}$ is a tuple. How is the tuple encoded into an embedding? How does the resulting $\mathcal{R}^u(q^*)$ work? What is the observation? How are personas and observations combined to perform personalization? What is the textual loss function? These are unclear.
2. The motivation for the proposed module is unclear, making this paper less like a technical report.
3. Are the baseline methods adapted to the dataset used, or do they use their generalized form? Furthermore, do they also use the tools used in this paper?
4. There is a lack of discussion on the space and time complexity of the algorithm. Each user needs to maintain a large amount of information, and test time alignment may introduce significant inference delays.

**Questions:**

Please refer to Weaknesses.

---

> ### Author Response · Authors · 2025-11-23
>
> > W1. The paper is poorly written, the method is obscure, and lacks necessary details. For example, the input to $f_{enc}$ is a tuple. How is the tuple encoded into an embedding? How does the resulting $\mathcal{R}^u(q^*)$ work? What is the observation? How are personas and observations combined to perform personalization? What is the textual loss function? These are unclear.
>
> **A1.** We thank the reviewer q71x for these helpful suggestions.
>
> For episodic memory construction, each **interaction tuple** is serialized into a natural-language text sequence and concatenated with a task-specific instruction prompt, forming the full textual input to the encoding model $f_{enc}$ BM25 (detailed in Section 3.1). During agent execution, the **observation** consists of either retrieved episodic memories or tool outputs (Wikipedia search results). This observation is then integrated with the persona (system prompt) and prior thinking process to guide the agent to determine the next action and generate a personalized final response.
>
> To further improve transparency, we added **Appendix A**, which provides a formal definition of the textual loss function based on the TextGrad framework [1], and **Appendix J**, which presents a complete, step-by-step agent execution trace. These additions make explicit how observations are formed, and how persona and observations are combined to produce personalized actions and responses.
>
> > W2. The motivation for the proposed module is unclear, making this paper less like a technical report.
>
> **A2.** We appreciate this comment. To strengthen the technical motivation, we revised the paper to explicitly frame our design around two fundamental limitations of prior work to achieve user personalization (summarized in Table 1):
>
> - **Long-horizon personal data utilization**, which is not well supported by existing workflow-based or general agent methods.
>
> - **Individual-level user preference alignment**, which typically requires infeasible computation resources and cannot achieve real-time personalization in prior work.
>
> To address these gaps, we introduce:
>
> - **A cognitively inspired memory system** (episodic + semantic) for scalable long-term personalization.
>
> - **A novel test-time user-preference alignment algorithm** that performs on-the-fly, user-specific preference optimization.
>
> We strengthened Method sections to more clearly connect these design choices with the shortcomings of existing approaches.

---

> ### Author Response · Authors · 2025-11-23
>
> > W3. Are the baseline methods adapted to the dataset used, or do they use their generalized form? Furthermore, do they also use the tools used in this paper?
>
> **A3.**  Thanks for the question. All baselines were faithfully adapted to the LaMP benchmark following their respective original papers.
>
> - **Prompting, RAG, and PAG** were implemented using the official paper experimental protocols.
>
> - For **agentic baselines** (ReAct, MemBank), we implemented them using a unified LangChain framework, ensuring consistent tool interfaces (same tools used in our methods).
>
> - For **MemBank**, we followed the original memory design, while adapting the stored memory units to LaMP-style interaction events to ensure fairness.
>
> We have clarified these implementation details in the experimental setup section (Appendix E).
>
>
> > W4. There is a lack of discussion on the space and time complexity of the algorithm. Each user needs to maintain a large amount of information, and test time alignment may introduce significant inference delays.
>
> **A4.** We appreciate this important point.
> Importantly, our test-time user-preference alignment is performed **asynchronously** and is conducted immediately after each session. This means it does not add latency to the next user session as it is ready in advance, and therefore does not affect online inference responsiveness.
> Moreover, we added empirical runtime comparisons demonstrating that PersonaAgent introduces only moderate overhead compared to lightweight workflows, and remains more efficient than other agentic frameworks.
>
> | Task / Avg runtime per sample (s)         |  PAG | ReAct | MemBank | PersonaAgent |
> |-------------------------------------------|------|-------|------------|--------------|
> | LaMP-1: Personalized Citation Identification | 0.84 | 1.95 | 2.35       | 1.42         |
> | LaMP-2M: Personalized Movie Tagging          | 1.46 | 2.92 | 3.10       | 1.62         |
> | LaMP-2N: Personalized News Categorization    | 1.08 | 2.48 | 2.90       | 2.12         |
> | LaMP-3: Personalized Product Rating          | 1.58 | 3.10 | 3.34       | 1.98         |
> | **Average**                                  | 1.24 | 2.61 | 2.92       | 1.79         |

---

> ### Author Response · Authors · 2025-11-26
> **Follow-up on Rebuttal**
>
> We sincerely thank the reviewer q71x for the constructive feedback, which has greatly helped us improve the clarity and quality of this work. We have carefully incorporated the suggestions in the revised manuscript. We would be happy to further address any remaining concerns or provide additional clarifications if needed.

---

### Official Review · Reviewer_acgQ · 2025-11-01

**Soundness:** 3
**Presentation:** 3
**Contribution:** 2
**Rating:** 4
**Confidence:** 3

**Summary:**

This paper proposes PersonaAgent, a personalized LLM agent framework for conversational AI. PersonaAgent integrates two memory types—episodic and semantic memory—and a personalized action module, all coordinated via a dynamically optimized user persona (system prompt). The framework introduces a test-time user-preference alignment strategy that updates the persona prompt based on recent user interactions. Experiments on the LaMP benchmark demonstrate improved performance over non-personalized, workflow-based, and agentic baselines.

**Strengths:**

* Proposes a unified memory-action framework for personalization, generalizable across tasks.

* Introduces a test-time persona optimization mechanism, enabling real-time adaptation to user preferences.

* Provides comprehensive experiments and ablation studies, showing the necessity of each component.

**Weaknesses:**

* Evaluation relies on machine metrics (accuracy, F1, ROUGE) not fully convincing; would be better to include personalization metrics (e.g., Persona-F1, faithfulness).

* The computational cost and scalability of test-time alignment are not thoroughly discussed.

**Questions:**

* How does the test-time alignment impact inference latency and scalability?

* How does the method perform for users with limited interaction history?

---

> ### Author Response · Authors · 2025-11-23
>
> > W1. Evaluation relies on machine metrics (accuracy, F1, ROUGE) not fully convincing; would be better to include personalization metrics (e.g., Persona-F1, faithfulness).
>
> **A1.**
> We appreciate the suggestion. Our current evaluation relies on task-grounded metrics (Accuracy, F1, ROUGE) because the ground-truth labels in LaMP are **real user behaviors and decisions, not synthetic data**. As such, task success directly reflects real-user satisfaction and decision alignment in practical settings. LaMP has become the most widely adopted benchmark in personalized LLM research, and all major follow-up works [1-4] use the same standardized metrics for fair comparison.
>
> Regarding Persona-F1 and faithfulness, they are used for evaluating how well a persona is expressed in persona-grounded response, and how factual consistency of generated text with respect to user features. However, these metrics inherently require ground-truth persona or user-profile features during evaluation, which are **not available** in our setting. Moreover, we **cannot compare** to strong baselines (e.g., RAG, ReAct) that do not construct or maintain explicit persona representations.
>
>
> These metrics primarily only assess **intermediate processes** in personalization tasks (given ground-truth persona/user features to evaluate faithfulness over model response), whereas our work focuses on **end-to-end personalization** (given only user raw history to evaluate user satisfaction over model response). Therefore, such metrics fall outside the scope of this paper. We believe that future benchmarks providing explicit, ground-truth user profiles would enable more meaningful persona-centric evaluation, and we view this as an important direction for follow-up work.
>
>
> [1] Jang, Joel, et al. "Personalized soups: Personalized large language model alignment via post-hoc parameter merging." arXiv preprint arXiv:2310.11564 (2023).
>
> [2] Tan, Zhaoxuan, et al. "Democratizing Large Language Models via Personalized Parameter-Efficient Fine-tuning." Proceedings of the 2024 Conference on Empirical Methods in Natural Language Processing. 2024.
>
> [3] Tan, Zhaoxuan, Zheyuan Liu, and Meng Jiang. "Personalized Pieces: Efficient Personalized Large Language Models through Collaborative Efforts." Proceedings of the 2024 Conference on Empirical Methods in Natural Language Processing. 2024.
>
> [4] Zhuang, Yuchen, et al. "Hydra: Model factorization framework for black-box llm personalization." Advances in Neural Information Processing Systems 37 (2024): 100783-100815.

---

> ### Author Response · Authors · 2025-11-23
>
> > W2. The computational cost and scalability of test-time alignment are not thoroughly discussed.
> Q1. How does the test-time alignment impact inference latency and scalability?
>
> **A2.** Thank you for raising this constructive point.
> Unlike approaches in Table 1 that require per-user fine-tuning of the underlying LLM, our method adopts a lightweight test-time preference optimization strategy over the agent persona. This design avoids expensive parameter updates and enables efficient scaling to large user populations in real-world online system.
>
> For the latency, our test-time user-preference alignment is designed to be **asynchronous**: it is performed immediately after each session and therefore does not introduce latency to the next user session. This design ensures that alignment improves the future quality of responses without blocking real-time usage. To quantify runtime overhead, we provide a direct comparison of average inference latency per sample with other baselines.
>
> | Task / Avg runtime per sample (s)         |  PAG | ReAct | MemBank | PersonaAgent |
> |-------------------------------------------|------|-------|------------|--------------|
> | LaMP-1: Personalized Citation Identification | 0.84 | 1.95 | 2.35       | 1.42         |
> | LaMP-2M: Personalized Movie Tagging          | 1.46 | 2.92 | 3.10       | 1.62         |
> | LaMP-2N: Personalized News Categorization    | 1.08 | 2.48 | 2.90       | 2.12         |
> | LaMP-3: Personalized Product Rating          | 1.58 | 3.10 | 3.34       | 1.98         |
> | **Average**                                  | 1.24 | 2.61 | 2.92       | 1.79         |
>
> These results show that PersonaAgent introduces only moderate overhead compared to lightweight workflows (PAG), while remaining consistently more efficient than other agentic frameworks (ReAct, MemBank).
>
> > W3 (Q2). How does the method perform for users with limited interaction history?
>
> **A3.** Great question. We explicitly evaluate cold-start users by resampling users with fewer than 10 historical interactions.
>
> | Dataset | Metric | PAG | ReAct | MemBank | PersonaAgent |
> |---------|--------|-----|-------|------------|--------------|
> | LaMP-1: Personalized Citation Identification | Acc ↑ | 0.772 | 0.802 | 0.789 | **0.845** |
> |  | F1 ↑ | 0.764 | 0.792 | 0.781 | **0.837** |
> | LaMP-2M: Personalized Movie Tagging | Acc ↑ | 0.392 | 0.428 | 0.415 | **0.476** |
> |  | F1 ↑ | 0.328 | 0.361 | 0.349 | **0.407** |
> | LaMP-2N: Personalized News Categorization | Acc ↑ | 0.678 | 0.714 | 0.698 | **0.756** |
> |  | F1 ↑ | 0.438 | 0.468 | 0.452 | **0.509** |
> | LaMP-3: Personalized Product Rating | MAE ↓ | 0.354 | 0.329 | 0.342 | **0.291** |
> |  | RMSE ↓ | 0.660 | 0.628 | 0.646 | **0.589** |
>
> PersonaAgent consistently remains the strongest model across all tasks, demonstrating its ability to effectively exploit sparse user signals via test-time alignment and persona-driven memory utilization.

---

> ### Author Response · Authors · 2025-11-26
> **Follow-up on Rebuttal**
>
> We sincerely thank the reviewer acgQ for the constructive feedback, which has greatly helped us improve the clarity and quality of this work. We would be happy to further address any remaining concerns or provide additional clarifications if needed.

---

### Official Review · Reviewer_dcVc · 2025-11-01

**Soundness:** 2
**Presentation:** 3
**Contribution:** 3
**Rating:** 4
**Confidence:** 3

**Summary:**

This paper proposes PersonaAgent, the first personalized LLM agent framework that adapts to individual user preferences through a dynamic persona. It combines personalized memory (episodic and semantic) and action modules, with the persona acting as an intermediary that evolves via user interactions. Experiments show it outperforms baselines in personalization and scales effectively in real-world test-time settings.

**Strengths:**

1.	PersonaAgent is the first LLM agent framework for dynamic user-level personalization, combining episodic/semantic memory and persona-driven actions for continuous adaptation.
2.	The test-time alignment method optimizes the persona via simulated interactions and textual loss, enabling real-time, scalable personalization without retraining.
3.	The work rigorously validates its approach across four diverse personalization tasks, ablation studies, and scaling analyses.

**Weaknesses:**

1.	The evaluation relies primarily on LaMP, which focuses on text classification and generation tasks that do not adequately capture instruction-following ability in real interactive dialogues—would the framework still excel other benchmarks?
2.	The action module only uses Wikipedia search and personal data retrieval; given that Wikipedia search may dominate performance gains, does the personalization component (i.e., personal data retrieval alone) meaningfully contribute to the agent’s effectiveness?
3.	Although persona case studies are included, the full agent execution process is not illustrated—could a detailed step-by-step example better demonstrate how personalization operates in practice?
4. The paper lacks runtime analysis—how long does the agent actually take to execute?

**Questions:**

See the weaknesses.

---

> ### Author Response · Authors · 2025-11-23
>
> > W1. The evaluation relies primarily on LaMP, which focuses on text classification and generation tasks that do not adequately capture instruction-following ability in real interactive dialogues—would the framework still excel other benchmarks?
>
> **A1.** We thank the reviewer dcVc for this thoughtful concern. LaMP is currently the **only** widely adopted personalization benchmark that provides **real user** data with longitudinal interaction histories in seven diverse personalization tasks, and has been extensively used by follow-up works in personalized LLM research [1-4]. To the best of our knowledge, there are **no publicly available, user-centric, long-term interactive dialogue datasets** that provide real user conversations with persistent identity and long-session multi-turn dialogue histories.
>
> In contrast, most existing dialogue-oriented benchmarks either rely on **LLM-synthesized users** (e.g., LoCoMo [5], PersonaChat [6]) or are limited to **short-session**, multi-turn conversations (e.g., PersonaChat [6], DailyDialog [7]). LaMP, by providing longitudinal, real-user behavioral traces, is therefore uniquely suited to evaluating user-centric preference modeling  rather than generic memory recall.
>
> More importantly, LaMP tasks already incorporate instruction-following behavior via structured task-specific prompts and multi-turn interactions, making them closer to interactive settings than conventional single-turn benchmarks.
> We have revised the paper (Appendix F) to clarify our motivation for using LaMP and to explicitly discuss broader, real-user interactive evaluation as an important future direction when such datasets become available.
>
>
>
> > W2. The action module only uses Wikipedia search and personal data retrieval; given that Wikipedia search may dominate performance gains, does the personalization component (i.e., personal data retrieval alone) meaningfully contribute to the agent’s effectiveness?
>
> **A2.**
> Thanks for raising this important point. Here are two examples of evidence to show the contribution of personal data retrieval:
>
> - **ReAct uses Wikipedia search.** Although strong agentic baselines such as ReAct also use the same Wikipedia search tool, they fall behind PersonaAgent (ReAct: 0.450 Acc. / 0.378 F1 vs. PersonaAgent: 0.513 Acc. / 0.424 F1, LaMP-2M in Table 2). This demonstrates that external factual retrieval alone does not dominate all observed gains.
>
> - **Ablation of w/o memory.** In addition, our ablation study also demonstrates that episodic memory retrieval is the key driver of performance: when the episodic memory module is removed, performance drops substantially (Accuracy: 0.769 drops to 0.646, F1: 0.483 drops to 0.388, LaMP-2N in Table 3)
>
> [1] Jang, Joel, et al. "Personalized soups: Personalized large language model alignment via post-hoc parameter merging." arXiv preprint arXiv:2310.11564 (2023).
>
> [2] Tan, Zhaoxuan, et al. "Democratizing Large Language Models via Personalized Parameter-Efficient Fine-tuning." Proceedings of the 2024 Conference on Empirical Methods in Natural Language Processing. 2024.
>
> [3] Tan, Zhaoxuan, Zheyuan Liu, and Meng Jiang. "Personalized Pieces: Efficient Personalized Large Language Models through Collaborative Efforts." Proceedings of the 2024 Conference on Empirical Methods in Natural Language Processing. 2024.
>
> [4] Zhuang, Yuchen, et al. "Hydra: Model factorization framework for black-box llm personalization." Advances in Neural Information Processing Systems 37 (2024): 100783-100815.
>
> [5] Maharana, Adyasha, et al. "Evaluating Very Long-Term Conversational Memory of LLM Agents." Proceedings of the 62nd Annual Meeting of the Association for Computational Linguistics (Volume 1: Long Papers). 2024.
>
> [6] Jandaghi, Pegah, et al. "Faithful persona-based conversational dataset generation with large language models." Proceedings of the 6th Workshop on NLP for Conversational AI (NLP4ConvAI 2024). 2024.
>
> [7] Li, Yanran, et al. "DailyDialog: A Manually Labelled Multi-turn Dialogue Dataset." Proceedings of the Eighth International Joint Conference on Natural Language Processing (Volume 1: Long Papers). 2017.

---

> ### Author Response · Authors · 2025-11-23
>
> > W3. Although persona case studies are included, the full agent execution process is not illustrated—could a detailed step-by-step example better demonstrate how personalization operates in practice?
>
> **A3.** We agree this would increase clarity. We have added a detailed example for the full-agent execution in the paper Appendix J.
>
> > W4. The paper lacks runtime analysis—how long does the agent actually take to execute?
>
> **A4.** We thank reviewer dcVc for raising this important point. We have added a runtime analysis comparing PersonaAgent with strong baselines:
> | Task / Avg runtime per sample (s)         |  PAG | ReAct | MemBank | PersonaAgent |
> |-------------------------------------------|------|-------|------------|--------------|
> | LaMP-1: Personalized Citation Identification | 0.84 | 1.95 | 2.35       | 1.42         |
> | LaMP-2M: Personalized Movie Tagging          | 1.46 | 2.92 | 3.10       | 1.62         |
> | LaMP-2N: Personalized News Categorization    | 1.08 | 2.48 | 2.90       | 2.12         |
> | LaMP-3: Personalized Product Rating          | 1.58 | 3.10 | 3.34       | 1.98         |
> | **Average**                                  | 1.24 | 2.61 | 2.92       | 1.79         |
>
>
> Although absolute latency varies with tasks and API conditions, the relative runtime ordering remains consistent: PAG < PersonaAgent < ReAct < MemBank. This demonstrates that our PersnoaAgent introduces only moderate overhead compared to workflow-based baselines (PAG), while remaining more efficient than other agentic frameworks (ReAct and MemBank). The user-aligned persona enables more direct and confident responses, whereas other agentic methods often incur additional latency due to inefficient user modeling and redundant function calls.

---

> ### Author Response · Authors · 2025-11-26
> **Follow-up on Rebuttal**
>
> We sincerely thank the reviewer dcVc for the constructive feedback, which has greatly helped us improve the clarity and quality of this work. We have carefully incorporated the suggestions in the revised manuscript. We would be happy to further address any remaining concerns or provide additional clarifications if needed.

---

### Note · Authors · 2026-01-06

I have read and agree with the venue's withdrawal policy on behalf of myself and my co-authors.